# *CYP51* Mutations in the *Fusarium solani* Species Complex: First Clue to Understand the Low Susceptibility to Azoles of the Genus *Fusarium*

**DOI:** 10.3390/jof8050533

**Published:** 2022-05-20

**Authors:** Pierre Vermeulen, Arnaud Gruez, Anne-Lyse Babin, Jean-Pol Frippiat, Marie Machouart, Anne Debourgogne

**Affiliations:** 1Laboratoire Stress Immunité Pathogènes, UR 7300, Faculté de Médecine, Université de Lorraine, 9 Avenue de la Forêt de Haye, F-54500 Vandœuvre-lès-Nancy, France; p.vermeulen@chru-nancy.fr (P.V.); anne-lyse.babin@univ-lorraine.fr (A.-L.B.); jean-pol.frippiat@univ-lorraine.fr (J.-P.F.); m.machouart@chru-nancy.fr (M.M.); 2Service de Microbiologie, CHRU de Nancy, Hôpitaux de Brabois, 11 Allée du Morvan, F-54511 Vandœuvre-lès-Nancy, France; 3IMoPA, CNRS, Université de Lorraine, F-54000 Nancy, France; arnaud.gruez@univ-lorraine.fr

**Keywords:** *Fusarium*, antifungal susceptibility, *CYP51*, azole

## Abstract

Members of *Fusarium solani* species complex (FSSC) are cosmopolitan filamentous fungi responsible for invasive fungal infections in immunocompromised patients. Despite the treatment recommendations, many strains show reduced sensitivity to voriconazole. The objective of this work was to investigate the potential relationship between azole susceptibility and mutations in CYP51 protein sequences. Minimal inhibitory concentrations (MICs) for azole antifungals have been determined using the CLSI (Clinical and Laboratory Standards Institute) microdilution method on a panel of clinical and environmental strains. *CYP51A*, *CYP51B* and *CYP51C* genes for each strain have been sequenced using the Sanger method. Amino acid substitutions described in multiple azole-resistant *Aspergillus fumigatus* (mtrAf) strains have been sought and compared with other *Fusarium* complexes’ strains. Our results show that FSSC exhibit point mutations similar to those described in mtrAf. Protein sequence alignments of CYP51A, CYP51B and CYP51C have highlighted different profiles based on sequence similarity. A link between voriconazole MICs and protein sequences was observed, suggesting that these mutations could be an explanation for the intrinsic azole resistance in the genus *Fusarium*. Thus, this innovative approach provided clues to understand low azole susceptibility in FSSC and may contribute to improving the treatment of FSSC infection.

## 1. Introduction

The genus *Fusarium* spp., mainly known as a plant pathogen, is cosmopolite and present in soil, water, and air. *Fusarium* is classified in different species complexes, and some of them have been described as human pathogens, such as *Fusarium solani* species complex (FSSC), *Fusarium oxysporum* species complex (FOSC), or *Fusarium fujikuroi* species complex (FFSC). Among them, the *Fusarium solani* species complex is the most represented [1,2]. Actually, the nomenclature of this complex is under revisions. Indeed, taxonomy experts propose to reclassify FSSC in a new genus, *Neocosmospora*. In contrast, the opposite view reasserts that FSSC has to be included in the genus *Fusarium*, being the most practical scientific option [3,4]. *Fusarium* can cause skin and ocular infections [5], but can also lead to severe invasive fungal infections, fusariosis, in immunodeficient patients, in particular those who have received stem cells transplants or suffer from acute leukemia [1,6,7]. The number of fusariosis cases has been steadily increasing over the last two decades [8,9].

Therapeutic drugs for fusariosis treatment are limited. Indeed, *Fusarium* spp. presents a low susceptibility to most antifungals. Due to the lack of clinical trials, the optimal treatment strategy for patients with fusariosis remains unclear. However, the European Society of Clinical Microbiology and Infectious Diseases (ESCMID) and the European Confederation of Medical Mycology (ECMM) published joint guidelines for the management of fusariosis and recommended the use of voriconazole, associated or not with amphotericin B [10,11]. Despite these recommendations, many studies have demonstrated a high variability of voriconazole MIC between *Fusarium* strains. In their article, Debourgogne et al. determined voriconazole MICs for 48 FSSCs from clinical and environmental samples by the CLSI M38-A2 method. They found a mean MIC of 4.6 µg/mL and a MIC_90_ of 8 µg/mL [12].

Voriconazole belongs to the class of azole antifungals, whose mechanism of action is based on the inhibition of ergosterol synthesis. Ergosterol is a major component of the fungal membrane. Its production is managed by the 14α-demethylase enzyme, the target of azole antifungals [13,14]. The binding of azole to the ferric iron moiety of the heme-binding site blocks the enzyme’s natural substrate lanosterol, disrupting the biosynthetic pathway [15]. In the *Fusarium* genus, this enzyme is encoded by the *CYP51* gene and is present in three isoforms: A, B, and C [16].

In fungi, azole resistance involves multiple mechanisms, such as (i) target alteration or overexpression, (ii) upregulation of multidrug transporters (efflux and impermeability), or (iii) cellular stress responses [17,18,19].

Qualitative modifications such as amino acid substitutions within the target inhibiting drug binding are a widespread azole-resistance mechanism in fungi. Protein overexpression is also a frequent mechanism among azole-resistant clinical isolates of yeasts, such as *C. albicans* with *ERG11* [19]. In filamentous fungi, the mechanism or the contribution of overexpression to azole resistance remains unclear [20]. Two cases of *CYP51A* overexpression in *A. fumigatus* have been described [21]. Another ubiquitous resistance mechanism involves membrane-associated efflux pumps. Two systems have been highlighted, the ATP-binding cassette (ABC) superfamily which use ATP hydrolysis, and the major facilitator superfamily (MFS) using the electrochemical proton-motive force to power drug efflux.

In *A. fumigatus*, target alteration inducing azole resistance can emerge in response to azole therapy [22]. The most commonly reported resistance mechanisms with CYP51A are substitutions at codons 54 and 220 [23]. However, resistance could also be of environmental origin and driven by the agricultural use of azoles. The first resistance mechanism described from environmental origin consists of a combination of a substitution at codon 98 in the *CYP51A* gene and a 34 base-pair tandem repeat (TR) in the gene promoter (TR_34_/L98H) [24].

The objective of this study was to identify a potential relationship between azole susceptibility and modification by target mutations in *Fusarium solani* species complex to better understand resistance mechanisms and help adapt treatment of this multi-resistant pathogen.

## 2. Materials and Methods

### 2.1. Isolates 

Seventeen FSSC isolates presenting different genotypes were used for this study [25]. Their CBS-KNAW Fungal Biodiversity Center collection numbers and characteristics are listed in Table 1. *Fusarium* strains were grown on Sabouraud broth (Sigma-Aldrich, St. Louis, MO, USA) agar medium and incubated at 30 °C. Strains were stored at −20 °C in water/glycerol (1/1).

### 2.2. Antifungal Susceptibility Testing

Minimal inhibitory concentration (MIC) determination by microdilution testing was performed following the CLSI M38-A2 reference method [26]. *Fusarium falciforme* (ATCC MYA-3636) was used as a quality control strain and was compliant for the study. Antifungals used in the study were voriconazole (VRZ) (range 0.03–16 µg/mL), itraconazole (ITZ) (range 0.03–16 µg/mL), posaconazole (PSZ) (range 0.03–16 µg/mL), and isavuconazole (IVZ) (range 0.03–16 µg/mL) (Sigma-Aldrich, St. Louis, MO, USA). MIC is the antifungal concentration that led to a complete inhibition of visual growth after 48 h of incubation at 37 °C. Each strain was tested in triplicate (biological replicate), and for each MIC determination, FSSC strains were tested in duplicate (technical replicate).

### 2.3. Genomic DNA Extraction

DNA was extracted from agar cultures using the Monarch Genomic DNA Purification Kit (New England Biolabs^®^ Inc., Ipswich, MA, USA) according to the manufacturer’s instructions. The DNA concentration was determined using a Nanodrop 2000 system (Thermo Scientific, Waltham, MA, USA).

### 2.4. Amplification and Sequencing of CYP51A, CYP51B, and CYP51C Genes

The *CYP51A*, *CYP51B,* and *CYP51C* genes were amplified using previously described primers [27]. The reaction mixtures contained 0.2 µM of each primer, 25 µL of Taq’Ozyme Purple Mix 2 (Ozyme, Saint-Cyr-l’École, France), and 5 µL of gDNA in a final volume of 50 µL. Samples were amplified using a CFX96 Real-Time PCR System (BioRad, Marnes-la-Coquette, France) and the following cycling protocol: one initial cycle of 2 min at 95 °C, followed by 35 cycles of 30 s at 95 °C, 30 s at 55 °C, and 2 min at 72 °C, with one final cycle of 5 min at 72 °C. PCR quality was assessed by 1.5% agarose gel electrophoresis (120 V, 20 min) and visualization using the Fusion Fx7 device (Vilber Lourmat, Eberhardzell, Germany). PCR product sequencing was performed using the Sanger method by Eurofins Genomics (Ebersberg, Germany) with the same primers.

### 2.5. Bioinformatic Analysis

Sequences and alignments have been interpreted using the Geneious Prime^®^ software (Geneious v2021.1.1 created by Biomatters, accessed in May 2021, available from https://www.geneious.com) and compared with a database using NCBI BLAST [28]. Nucleotide sequences were confirmed thanks to described sequences from *Fusarium vanettenii* mpVI 77-13-4 (accession numbers in Table 2). For each strain, mRNA sequences were obtained by sequence alignment with *CYP51A*, *CYP51B*, and *CYP51C* mRNAs of *Fusarium vanettenii* mpVI 77-13-4 (accession numbers in Table 2). Translation and protein sequence alignments were performed using Geneious Prime^®^ software. In this study, different protein profiles were identified, and two proteins were considered to belong to the same profile when the percentage of similarity was ≥99%.

### 2.6. Comparison of CYP Proteins’ Mutations

The positions of mutations in CYP51A and CYP51B were compared to those associated with azole resistance in *Aspergillus fumigatus* NRRL:163 (WT reference strain) [23]. Unfortunately, *A. fumigatus* does not possess a third CYP51 isoform. However, CYP51C has been characterized in *Aspergillus flavus* NRRL3357 (WT reference strain) and mutations associated with azole resistance have been described despite controversial data in the literature. Liu et al. showed that a T788G mutation in the *CYP51C* gene was responsible for voriconazole resistance in *A. flavus*. However, Paul et al. contradicted this conclusion because this mutation was not found in their strains and another mutation (Y319H) has been highlighted and implicated in mediating voriconazole resistance [29,30,31]. CYP51A protein sequences from our panel of FSSC strains have been compared to other complexes in the genus *Fusarium*. Reference strains from environmental and clinical *Fusarium* complex species known to be pathogenic in humans or plants have been selected (Table 3). This comparison was carried out to highlight shared point mutations that could explain voriconazole susceptibility variability at the complex or genus levels.

### 2.7. Protein Modeling

The three-dimensional (3D) homology structure of FSSC CYP51A protein was modeled by using the I-TASSER server (http://zhanglab.ccmb.med.umich.edu/I-TASSER (accessed in October 2021)) [32,33], using *Fusarium vanettenii* mpVI 77-13-4 as a query sequence. The crystal structure of lanosterol 14-alpha demethylase (PDB ID 5eqb), sharing a sequence identity of 45.1% and a TM-score of 0.984 (coverage of 99%), was used as the template for model building. The CYP51A model was studied and figures were prepared using the PyMOL Molecular Graphics System, v2.5.2 (Schrödinger, LLC, New York, NY, USA).

## 3. Results

### 3.1. Antifungal Susceptibility Testing

Voriconazole was the antifungal agent with high MIC variabilities between strains. Its MIC median and ranges were, respectively, 8 and 2–16 µg/mL. Eight strains among seventeen (47%) had a VRZ MIC value of 8 µg/mL. All strains presented high MICs for itraconazole and posaconazole, with values higher than 16 µg/mL, except FSSC20 which presented a MIC of posaconazole of 8 µg/mL. Determining the exact MIC value over 16 µg/mL for ITZ and PSZ was not possible due to the lack of solubility of the azole agents. Like voriconazole, isavuconazole presented variability of susceptibility. Its MIC median was 16 µg/mL (range 4–64 µg/mL) (Table 4). MICs for the quality control strain were in the expected range for each experiment.

### 3.2. Protein Profiles and Minimal Inhibitory Concentration

Protein sequence analysis of the 17 FSSC strains revealed 54, 23, and 29 mutations for CYP51A, B, and C, respectively. The comparison of protein sequences highlighted 9 profiles for CYP51A (1a to 9a), 4 profiles for CYP51B (1b to 4b), and 5 profiles for CYP51C (1c to 5c) (Table 5). The distribution of the different FSSC strains according to their voriconazole MICs revealed a link between MIC and protein sequence. Indeed, strains with low voriconazole MICs shared the same profile, except for FSSC20 and FSSC99, which presented a different VRZ MIC from one dilution.

### 3.3. Comparison with Aspergillus fumigatus CYP51A

Only 5 of the 23 mutations (22, 172, 220, 255, and 427) described in *A. fumigatus* as being associated with azole resistance were found in our FSSC strains (Table 6). Mutations at positions 22, 220, 255, and 427 in FSSC strains were different than those of *A. fumigatus* wild-type (*Af* WT) and mutations observed at these positions in *A. fumigatus* resistant to azoles (*Af* R). On the contrary, the mutation at position 172 was different from *Af* WT but identical to *Af* R. All FSSC strains shared these mutations, inducing the same amino acid changes.

### 3.4. Modeling Structure of FSSC CYP51A Protein

The five mutations shared by all FSSC strains and involved in azole resistance of *A. fumigatus* have been highlighted in the model of CYP51A in order to assess their impact on 14α-demethylase (Figure 1). The amino acids at positions 22, 170, 253, and 422 are not located near the active site nor near the substrate channel, but at the protein surface. The side chain of H22 interacts with a crevice formed by residues W41, L42, and P43 that are located in the loop before the short αA′ helix according to CYP nomenclature. This helix participates at the channel entry of the substrate. As a consequence, the mutation at position 22 can influence the positioning of the αA′ helix and reshape the surface of the channel entry. This modification could induce azole discrimination or forbid the accessibility of the azole to the active site by steric hindrance, depending on the interaction that could block the entry of the active site against the membrane. Concerning the point mutations at positions 170, 253, and 422, they are located in the β31, αG-αH, and αK′-αL loops, respectively. How these three substitutions might impact azole binding is unknown and will require further study. Amino acid L218 interacts with amino acid F75, and both of them are located at the entry of the substrate channel of CYP51A, in close proximity to the modeled itraconazole. As a consequence, position 218 might be implicated in the interaction between the enzyme and its substrates. Physicochemical properties of the amino acids observed in the five mutations of interest (position 22, 170, 218, 253, and 422 in FSSC strains) have been compared with those described in *A. fumigatus* (Table 7). As described in the literature, mutations of CYP51A M220 to I, K, T, or V residues were associated with azole resistance in *A. fumigatus*. Interestingly, in FSSC strains, the amino acid at the equivalent position is a leucine (L218). This residue is structurally very close to isoleucine, which is present in all FSSC strains.

### 3.5. Comparison of Aspergillus fumigatus CYP51B and Aspergillus flavus CYP51C

The same process has been applied for CYP51B. None of the four mutations described in *A. fumigatus* have been observed in the panel of studied FSSC strains. At these positions, FSSC strains share the same amino acid as the one described in the WT *A. fumigatus* reference strain.

Only two mutations in CYP51C have been described and associated with *Aspergillus flavus* resistance to azoles. At position 240 (249 for FSSC), all FSSC strains present a different amino acid (G) than *A. flavus* WT (S) and azole resistant (A). Concerning the second mutation, FSSC strains share the same amino acid as the one described in the WT *A. flavus* reference strain

### 3.6. Comparison of FSSC vs. Other Fusarium Species Complex

We have demonstrated that all studied FSSC strains share the same amino acid mutations known to be associated with azole resistance in *A. fumigatus*. We focused on the five mutations in CYP51A sequences described previously (H22, V170, L218, G253, D422) and compared them with other FSSC genotypes as well as different species complexes previously described in the literature (*Fusarium oxysporum* species complex (FOSC), *Fusarium fujikuroi* species complex (FFSC), and *Fusarium sambucinum* species complex (FSAMSC)) (Table 8). Other genotypes belonging to FSSC present the same amino acids as our panel. Interestingly, FSAMSC differs from our panel, with only two common amino acids among the five point mutations. The H22 mutation seems to be specific to FSSC as only FSSC strains share this mutation, which is substituted by an asparagine (N) in the other species complexes. Concerning position 218 located at the channel entry, all the strains from the different complexes present the same amino acid (leucine (L)).

## 4. Discussion

MIC determination in our panel of FSSC strains showed a low susceptibility for azoles used in clinical practice and a high variability of susceptibility for voriconazole and isavuconazole (MIC values ranged from 2 to 16 and 4 to 64 µg/mL, respectively). Our MIC values were close to the range of 1–16 µg/mL formerly defined for *Fusarium solani* complex species [1,35,36]. Our results also revealed that all FSSC strains exhibited itraconazole and posaconazole MICs superior to 16 µg/mL, except for the FSSC20 strain which had a MIC_PSZ_ of 8 µg/mL. This observation has also been reported in other publications [10,35]. Tortorano et al. showed comparable results for posaconazole as the majority of their FSSC strains presented MICs of 16 µg/mL, except for one strain for which it was 1 µg/mL [11]. Concerning isavuconazole, the results showed a larger range of MIC values, from 4 to 64 µg/mL. Most of its activities against *Fusarium* spp. have shown MIC values often greater than 16 or even 32 µg/mL [10,37,38,39]. Our findings support the high MIC values previously reported and revealed that 64% of FSSC strains exhibited MICs superior or equal to 16 µg/mL.

The variability of susceptibility to voriconazole observed for FSSC strains in our study raises the question of the mechanism responsible for this low susceptibility. Alignments of CYP51A, CYP51B, and CYP51C protein sequences highlighted similarities between strains, but also allowed their classification into different profiles: 9, 4, and 5 profiles for CYP51A, CYP51B, and CYP51C, respectively. To the best of our knowledge, this approach has not been carried out before. Furthermore, sequence profiling allowed to link voriconazole MICs with the CYP51 protein sequence.

Up to now, no mutations involved in azole resistance within the 14α-demethylase gene of the *Fusarium solani* species complex had been described. By comparing with CYP51A of *Aspergillus fumigatus*, we showed that 5 of the 23 mutations described in the literature as being associated with a change in azole sensitivity in this pathogen were found in all tested FSSC strains. Most of these mutations have been described as responsible for voriconazole resistance in *A. fumigatus* [23]. In this fungus, a substitution of methionine 220 to lysine, isoleucine, valine, or threonine impacts voriconazole susceptibility. Indeed, it has been shown that clinical strains exhibiting the M220I mutation present increased voriconazole MICs [40]. This fact has also been proven using yeast clones expressing the CYP51 M220I mutant of *A. fumigatus* [41]. Isoleucine found in *A. fumigatus* is very structurally close to leucine found at position 218 in our panel of FSSCs. They are both apolar amino acids and share the same physicochemical properties. It has been shown that changes in amino acids due to their physicochemical properties (polarity, hydrophobicity, etc.) are responsible for modifying protein conformation and consequently, its affinity with targets such as azoles [42]. Interestingly, L218, corresponding to M220 for *A. fumigatus,* is common to all strains. By structure modeling of CYP51A, with *Fusarium vanettenii* mpVI 77-13-4 as a query sequence, we observed that L218 is located at the channel entry of the substrate. This conserved leucine at position 218 and its role in protein channel conformation were also described by James et al. [27]. Thus, we propose that the L218 residue is directly involved in the discrimination of azole by modifying the shape of the channel entry of the substrate for the *Fusarium solani* species complex or associated with the genus *Fusarium*. Concerning the influence of the mutation at the position 22 on azole discrimination, one possibility could involve its influence on the orientation of the αA′ helix that constitutes an integral part of the channel entry, while the second possibility could be the locking of CYP51A against the membrane, that could infer with the accessibility of the long azole chain to the active site. Mutations at positions 170, 253, and 422 also led to a low susceptibility for azole in our study, whereas they are located far from the azole binding site. Albeit their precise role in azole discrimination is difficult to assess at the molecular level, one can hypothesize that amino acids at these positions could be key residues that affect the plasticity of CYP51A within *Fusarium solani* species complex, which is required for ligand binding.

The literature reports that some strains of *A. fumigatus* show resistance or reduced sensitivity to one or more azoles without any mutations detected in the *CYP51A* gene. These phenotypes have been explained by the presence of mutations in the *CYP51B* gene [43,44]. These point mutations were not found in our panel. The same observation was made for *CYP51C* in comparison with *Aspergillus flavus.*

Few studies have focused on *Fusarium* species complexes other than FSSC and their resistance to clinically used azoles. Indeed, the genus *Fusarium* is also a plant pathogen. Studies about *Fusarium graminearum* (FSAMSC) and FOSC have shown that the *CYP51A* gene encodes an inducible 14α-demethylase, determining sensitivity to azoles. On the contrary, CYP51B is a 14α-demethylase involved in the formation of ascospores, but is barely impacted by antifungals. Indeed, an increase in sensitivity to azole antifungals has been described in strains deleted in *CYP51A* (Δ*CYP51A*), but no difference was observed for Δ*CYP51B* mutants. *CYP51C* is genus-specific and is found in the genus *Fusarium*. However, its role is not clearly established. Studies show that *CYP51C* is not involved in sterol 14-demethylation, but gene deletion increases in sensitivity for some sterol demethylase inhibitors [45,46]. Recently, novel *CYP51* paralogues have been described. Similar to *CYP51B*, a new paralogue named *CYP51D* is involved in azole resistance and occurs in genomes of fungi belonging to Eurotiomycetes. A novel partial *CYP51A* gene encodes a truncated form of CYP51A responsible for azole resistance [47]. Therefore, we focused on CYP51A, the protein most responsible of azole resistance in the genus *Fusarium* [20,45,48,49,50]. To further characterize our findings about amino acid changes, we compared the CYP51A protein sequence of our FSSC strains with other strains from the same complex and with strains from different complexes. All strains from the different complexes presented the same amino acid at position 218 (L218) as that in our panel. Our hypothesis is that this observation could be a first clue to explain the low susceptibility of clinically used azole drugs in the genus *Fusarium.* More studies should be performed to confirm and generalize our findings in fusarioid fungi.

## Figures and Tables

**Figure 1 jof-08-00533-f001:**
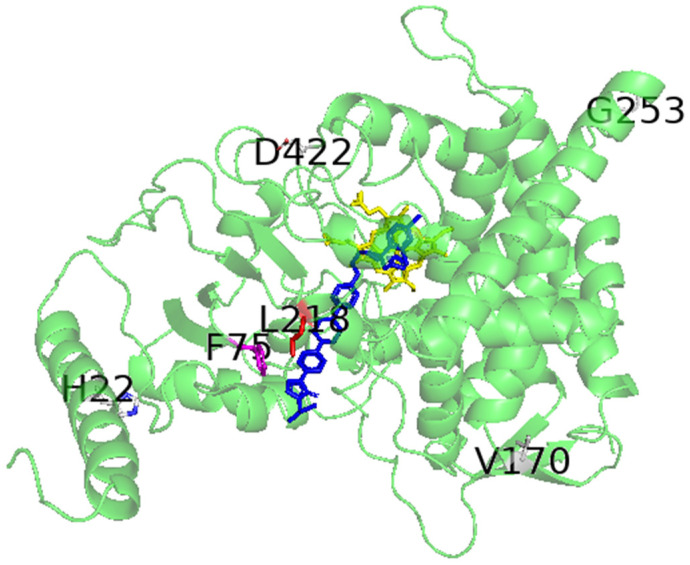
Calculated model of FSSC CYP51A protein. Point mutations shared by all FSSC strains and known to be associated with azole resistance in *A. fumigatus* are indicated by the amino acid letter and position. Amino acid L218 interacts with F75, and both are present at the entry of the substrate channel (yellow: heme; blue: itraconazole; purple: F75; red: L218, corresponding to the channel entry for the substrate).

**Table 1 jof-08-00533-t001:** Characteristics of the *Fusarium solani* species complex strains used in this study.

Study Number	Collection Number	Genotype	Origin		
			Location	Geography	Year of Isolation
FSSC20	CBS 124901	2-w	Skin	France	2008
FSSC21	CBS 124631	1-b	Nail	France	2008
FSCC25	CBS 124895	5-o	Skin	France	2008
FSSC29	CBS 124889	2-d	Nail	France	2008
FSSC36	CBS 124890	5-v	Nail	France	2008
FSSC45	CBS 124898	2-d	Skin	Gabon	2008
FSSC47	CBS 124892	20-f	Nail	Gabon	2008
FSSC86	CBS 102824	25-d	Plated litter fragment	Colombia	2000
FSSC90	CBS 208.29	25-d	*Hyacinthus orientalis*	Germany	no data
FSSC98	CBS 119996	5-kk	Daucus carota	Netherlands	no data
FSSC99	CBS 119223	21-f	Carrot	Spain	no data
FSSC102	CBS 115659	21-c	Potato cultivar Maritta	Germany	no data
FSSC111	CBS 115660	21-d	Potato	Egypt	no data
FSSC112	CBS 115658	21-e	Potato	Israel	no data
FSSC115	CBS 109028	28-c	Subcutaneous nodule	Switzerland	no data
FSSC118	CBS 224.34	1-b	Toe nail	Cuba	1929
FSSC121	CBS 117608	6-f	Arm lesion human dermis	Turkey	no data

**Table 2 jof-08-00533-t002:** Genes and mRNA accession numbers of *Fusarium vanettenii* mpVI 77-13-4 used in this study.

Isoform	Gene	mRNA
*CYP51A*	NECHADRAFT_43488 (1583 bp)	XM_003045158.1 (1521 bp)
*CYP51B*	NECHADRAFT_57370 (1965 bp)	XM_003054190.1 (1802 bp)
*CYP51C*	NECHADRAFT_41888 (1649 bp)	XM_003051375.1 (1551 bp)

**Table 3 jof-08-00533-t003:** *Aspergillus* sp. and *Fusarium* complex references and accession numbers of CYP51A, CYP51B, and CYP51C protein sequences used in our comparisons.

Protein	Complex	Denomination	Accession Number
CYP51A	/	*Aspergillus fumigatus* NRRL:163	AAK73659
	*Fusarium solani* complex	*Fusarium falciforme*	QGZ00344.1
		*Fusarium floridanum*	RSL78044.1
		*Fusarium keratoplasticum*	QGR26268.1
		*Fusarium suttonianum*	QGZ00346.1
	*Fusarium oxysporum* complex	*Fusarium odoratissimum* NRRL 54006	XP_031067647.1
		*Fusarium oxysporum* f. sp. *lycopersicum* 4287	XP_018249823.1
	*Fusarium fujikuroi* complex	*Fusarium denticulatum*	KAF5688222.1
		*Fusarium fujikuroi* IMI 58289	XP_023435732.1
		*Fusarium proliferatum* ET1	XP_031088172.1
		*Fusarium verticillioides* 7600	XP_018757407
	*Fusarium sambucinum* complex	*Fusarium graminearum* PH-1	XP_011321548.1
		*Fusarium pseudograminearum* CS3096	XP_009251504.1
CYP51B	/	*Aspergillus fumigatus* NRRL:163	AAK73660
CYP51C	/	*Aspergillus flavus* NRRL3357	QRD94494

**Table 4 jof-08-00533-t004:** Antifungal susceptibility results of FSSC strains: MIC median and ranges.

Antifungal	MIC (µg/mL)	
	Median	Range
Voriconazole	8	2–16
Itraconazole	>16	/
Posaconazole	>16	/
Isavuconazole	16	4–64

**Table 5 jof-08-00533-t005:** Minimum inhibitory concentrations (MICs) of the FSSC strains for azole antifungals (voriconazole (VRZ), itraconazole (ITZ), posaconazole (PSZ), isavuconazole (IVZ)) and corresponding homology profiles of CYP51A, B, and C for each FSSC strain (n/a: data unavailable). Studied strains are classified in ascending order of voriconazole MICs. MICs for other tested azoles are also shown. Each protein sequence of CYP51A, B, and C was compared between the different FSSC strains and observed homologies were used to define profiles. Two strains share the same profile when their protein sequences present more than 99% of similarity. Profiles are noted with a number and a letter corresponding to the isoform. Numbers were chosen arbitrarily based on the order of strains with respect to voriconazole MIC. For each CYP51 isoform, the color gradient corresponds to the proximity of the protein profiles to each other (from green to red).

Strains	Antifungal MICs (µg/mL)				Profiles		
	VRZ	ITZ	PSZ	IVZ	CYP51A	CYP51B	CYP51C
FSSC20	2	>16	8	16	3a	2b	2c
FSSC102	2	>16	>16	8	1a	1b	n/a
FSSC111	2	>16	>16	4	1a	1b	1c
FSSC112	2	>16	>16	4	1a	n/a	1c
FSSC45	4	>16	>16	16	2a	2b	2c
FSSC98	4	>16	>16	16	4a	3b	3c
FSSC99	4	>16	>16	8	1a	1b	1c
FSSC115	4	>16	>16	4	7a	4b	5c
FSSC21	8	>16	>16	16	5a	4b	n/a
FSSC25	8	>16	>16	16	8a	4b	4c
FSSC29	8	>16	>16	16	2a	4b	2c
FSSC36	8	>16	>16	32	4a	3b	3c
FSSC86	8	>16	>16	16	6a	4b	4c
FSSC90	8	>16	>16	8	4a	3b	n/a
FSSC118	8	>16	>16	16	5a	4b	4c
FSSC121	8	>16	>16	16	8a	4b	4c
FSSC47	16	>16	>16	64	9a	3b	3c

**Table 6 jof-08-00533-t006:** Comparison of amino acid changes observed in our FSSC strains with those associated with azole resistance in *A. fumigatus* CYP51A (green: point mutations different from wild-type (WT) and mutated (azole R) *A. fumigatus*; orange: point mutation different from WT *A. fumigatus* but identical to resistant *A. fumigatus* (azole R)).

*Aspergillus fumigatus*			*Fusarium solani* Species Complex		
Position	Amino Acid		Position	Amino Acid	Strains
	WT	Azole R			
22	N	D	22	H	All
46	F	Y	46	F	All
52	S	T	52	S	All
54	G	E, K, R, V, W	54	G	All
98	L	H	98	L	All
121	Y	F	121	Y	All
138	G	C, R	138	G	All
141	Q	H	141	Q	All
147	H	Y	147	H	All
172	M	V	170	V	All
216	P	L	214	P	All
220	M	K, I, T, V	218	L	All
248	N	T	246	N	All
255	D	E	253	G	All
289	T	A	287	T	All
297	S	T	295	S	All
394	P	L	389	P	All
427	E	G, K	422	D	All
431	Y	C	426	Y	All
434	G	C	429	G	All
440	T	A	435	T	All
448	G	S	443	G	All
491	Y	H	486	Y	All
495	F	I	490	F	All

**Table 7 jof-08-00533-t007:** Mutations of interest observed in FSSC strains, corresponding amino acids, physicochemical properties, and their impact on azole resistance (A: apolar; P: polar uncharged, −: negative polar; +: positive polar; ATU: area of technical uncertainty, R: resistant, n/a: data unavailable).

*Aspergillus fumigatus*						*Fusarium solani* Species Complex	
Position	WT	Mutant	Azole Resistance [23,34]			Position	Amino Acid
			VRZ	ITZ	PSZ		
22	Asn (P)	Asp (−)	n/a	R	n/a	22	His (+)
172	Met (A)	Val (A)	R	R	V	170	Val (A)
220	Met (A)	Lys (+)	ATU/R	R	R	218	Leu (A)
		Ile/Va (A)					
		Thr (P)					
255	Asp (−)	Glu (−)	R	R	ATU	253	Gly (A)
427	Glu (−)	Gly (A)	R	R	V	422	Asp (−)
		Lys (+)					

**Table 8 jof-08-00533-t008:** Comparison of the five mutations of interest in CYP51A between studied strains and other strains from different species complexes (FSSC, FOSC, FFSC, FSAMSC) (D: aspartic acid; E: glutamic acid; G: glycine; H: histidine; I: isoleucine; L: leucine; N: asparagine; V: valine).

Complexes	Position				
	22	170	218	253	422
***Fusarium solani* species complex (FSSC)**					
Strains from this study	H	V	L	G	D
*Fusarium falciforme*	H	V	L	G	D
*Fusarium floridanum*	H	V	L	G	D
*Fusarium keratoplasticum*	H	V	L	G	D
*Fusarium suttonianum*	H	V	L	G	D
***Fusarium oxysporum* species complex (FOSC)**					
*Fusarium odoratissimum* NRRL 54006	N	V	L	G	D
*Fusarium oxysporum* f. sp. *lycopersicum* 4287	N	V	L	G	D
***Fusarium fujikuroi* species complex (FFSC)**					
*Fusarium denticulatum*	N	V	L	G	E
*Fusarium fujikuroi* IMI 58289	N	V	L	G	E
*Fusarium proliferatum* ET1	N	V	L	G	E
*Fusarium verticillioides* 7600	N	V	L	G	E
***Fusarium sambucinum* species complex (FSAMSC)**					
*Fusarium graminearum* PH-1	N	I	L	I	D
*Fusarium pseudograminearum* CS3096	N	I	L	I	D

## Data Availability

Not applicable.

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
