# Peer review of "CYP51 Mutations in the Fusarium solani Species Complex: First Clue to Understand the Low Susceptibility to Azoles of the Genus Fusarium"

_jof, 2022, doi:10.3390/jof8050533_

Round 1

Reviewer 1 Report

Few suggestions:

FSSC is currently under revision (see for discussion Geiser et al Phytopathology 2021 and Crous et al Stud Mycol 2021 for the competing approaches)

PLease consider the taxonomic compleixity and debate in the manuscript

Table 1: if available indicate also the year of isolation. THe level of sensitivity to the fungicide can also be discussed in the light of the age of the strains

"despite controversial data in the literature [27–29]"   please elaborate

In discussion authors state " All strains from the different complexes presented the same amino acid at position 218 (L218) than in our panel. This observation could be a first clue to explain the low susceptibility of clinically used azole drugs in the genus Fusarium."

It would be appropriate to discuss litterature that already used medical azoles on different Fusarium to discuss if results obtained in this study can be generalized. Morever it would be imporntat also to see data on Neocosmospora to appropriately address the data on fusarioid fungi.

Reviewer 2 Report

The manuscript is important for this edition of the Journal.  The text needs modifying in the light of existing literature.  CYP51C is genus specific and not involved in sterol 14-demethylation, the sequence heterogeneity is not involved in resistance. The second paper references this and discusses intrinsic resistance in the light of different CYP51s including CYP51D.

Fan et al 2013 New Phytologist 198 821-835.

Van Rhijn et al 2021 mBio e0184521
